

# An End-to-End autonomous driving model based on visual perception for temporary roads

Qinghua Su, Min Xie, Liyong Wang, Yue Song, Ao Cui and Zhihao Xie

Key Laboratory of Modern Measurement and Control Technology, Ministry of Education, Beijing Information Science and Technology University, Beijing, China

## ABSTRACT

**Background:** The research on autonomous driving using deep learning has made significant progress on structured roads, but there has been limited research on temporary roads. The End-to-End autonomous driving model is highly integrated, allowing for the direct translation of input data into desired driving actions. This method eliminates inter-module coupling, thereby enhancing the safety and stability of autonomous vehicles.

**Methods:** Therefore, we propose a novel End-to-End model for autonomous driving on temporary roads specifically designed for mobile robots. The model takes three road images as input, extracts image features using the Global Context Vision Transformer (GCViT) network, plans local paths through a Transformer network and a gated recurrent unit (GRU) network, and finally outputs the steering angle through a control model to manage the automatic tracking of unmanned ground vehicles. To verify the model performance, both simulation tests and field tests were conducted.

**Results:** The experimental results demonstrate that our End-to-End model accurately identifies temporary roads. The trajectory planning time for a single frame is approximately 100 ms, while the average trajectory deviation is 0.689 m. This performance meets the real-time processing requirements for low-speed vehicles, enabling unmanned vehicles to execute tracking tasks in temporary road environments.

# INTRODUCTION

In recent years, autonomous driving has become a popular research topic in the automotive industry, and autonomous driving technology has made rapid advancements. The advantage of traditional autonomous driving systems lies in their clear functional partitioning and system interpretability. However, there is also a risk that the system may crash when certain subsystems fail. The deep neural network-based autonomous driving methods have achieved tremendous success in the field of autonomous driving due to their high accuracy, strong robustness, and low cost. Compared to the success obtained by traditional modular approaches, End-to-End autonomous driving systems utilize sensor data as input and train deep neural networks to output control information. Therefore,

Corresponding author
Liyong Wang, wly_bistu@126.com

directly mapping sensor data to decisions can effectively reduce the uncertainty transfer between modules.

Among various perception sensors, camera requires low price and high data density compared to other sensors. Utilizing cameras instead of radar can effectively help control costs. Currently, typical deep learning techniques, such as convolutional neural networks, are widely used in image processing applications and are particularly suitable for autonomous driving. The End-to-End model in this study also uses images as data input to control the vehicle's movement based on temporary road information.

In terms of scenarios, traffic roads can be categorized into structured roads and unstructured roads. Compared to structured roads, unstructured roads present relatively fewer scenarios during normal driving. However, when these scenarios do occur, it often indicates an accident or traffic control situation, which conventional autonomous driving algorithms may struggle to navigate smoothly, such as in temporary roads constructed with traffic cones. Therefore, the research on autonomous driving algorithms for temporary road scenarios is of great significance for enhancing driving safety.

Therefore, the main content of this research includes: (1) developing an End-to-End autonomous driving model for unmanned vehicles on temporary roads; (2) creating temporary road scene maps and collecting expert training datasets using the CARLA simulation system; (3) model training and fine-tuning; (4) validating the model performance through simulation tests and field tests.

## RELATED WORK

### End-to-End autonomous driving systems

*Pomerleau (1988)* from Carnegie Mellon University utilized a single hidden layer fully connected network to train with the available driving samples, resulting in the first End-to-End driving system, which also marks the origin of End-to-End autonomous driving systems. NVIDIA proposed the PilotNet, which takes images as input and outputs vehicle steering control (*Bojarski et al., 2016*). Its network can autonomously recognize road edges and vehicles, demonstrating the feasibility of End-to-End systems. This method directly establishes a function mapping from the input image to the angle control, but the abstraction of the output is too low, resulting in low interpretability. *Wu et al. (2022)* proposed an End-to-End Trajectory guided Control Prediction (TCP) model that uses only one RGB image as input and outputs control trajectories. This model predicts both the planned trajectory and the vehicle control signal. *Chen & Krähenbühl (2022)* proposed the Learning from All Vehicles (LAV) model, which integrates information from its own vehicle and surrounding vehicles in network to make output decision. *Prakash, Chitta & Geiger (2021)* proposed the TransFuser model, which introduces global contextual reasoning to adapt to complex scenes, such as managing traffic coming from multiple directions at uncontrolled intersections. Compared to traditional path planning algorithms, End-to-End networks not only reduce the need for extensive manual rule design but also demonstrate a stronger ability to adapt to new environments and greater robustness.

## Transformer model in computer vision

The Transformer model first achieved great success in the field of natural language processing (NLP) and was then applied to the field of computer vision (*Vaswani et al., 2017*). Convolutional Vision Transformer (ViT) generates tokens through convolutional networks based on ViT, combining the advantages of both convolutional networks and Transformer networks (*Dosovitskiy et al., 2017*; *Wu et al., 2021*). The Swin Transformer network divides the image into individual windows, restricts the Transformer calculations to within these windows, and enhances the interaction between windows through shifted window attention (*Liu et al., 2021*). The Global Context Vision Transformer (GCViT) integrates the global context self-attention module and the local self-attention module by computing attention masks and shifting local windows. This method models both long-range and short-range spatial interactions to achieve enhanced image feature extraction (*Hatamizadeh et al., 2023*).

The Transformer is also widely used in the field of modality fusion. The TransformerFusion network employs Transformers to fuse sequential images and accomplish 3D scene reconstruction (*Bozic et al., 2021*). TransFuser and InterFuser integrate image and LiDAR data using transformer modules to generate local paths (*Prakash, Chitta & Geiger, 2021*; *Shao et al., 2023*).

## MODEL CONSTRUCTION

As shown in Fig. 1, the End-to-End autonomous driving model proposed in this research consists of five parts: (1) Sensor Data Input Module. Three cameras are used as the road information collectors, and the captured images will be processed for further usages. (2) Image Feature Extraction Module. It uses GCViT as the backbone network to extract intermediate features of RGB images. (3) The Transformer Fusion Module. It achieves the fusion encoding of image features and has different outputs during the training phase and deployment phase. (4) The Local Path Prediction Module. It predicts local paths instead of directly outputting vehicle control information, which reduces dependence on vehicle hardware while enhancing applicability to other platforms. (5) The controller. It takes the path as input and outputs appropriate angles for vehicle controlling.

## Sensor data input module

The Sensor Data Input Module captures road image data from three RGB cameras. The camera in the middle is facing straight ahead, and the angles between the left, middle, and right cameras are 60 degrees. The camera outputs road images at a rate of 25 frames per second, and the original image resolution is 1,280 × 720. Then, the image resolution is converted to 224 × 224 and Image Set $I$ ($I = \left\{ I_{left}, I_{front}, I_{right} \right\}$, $I_{left,front,right} \in \mathbb{R}^{3*W*H}$) is constructed, which is subsequently input into the Image Feature Extraction Module.

## Image feature extraction module

The Image Feature Extraction Module is based on the 'xtiny' version of the GCViT network, as illustrated in Fig. 2. This module comprises four GCViT blocks and subsequently outputs the extracted image features $F$. Each GCViT block is composed of a

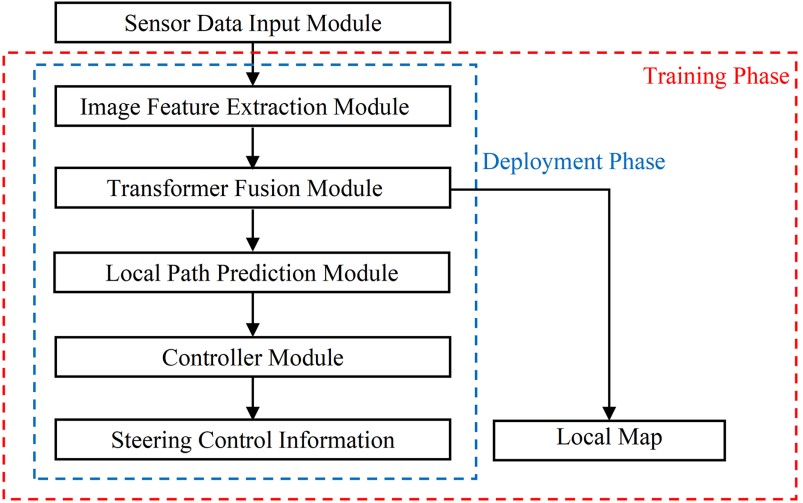

**Figure 1** The structure diagram of End-to-End autonomous driving model.

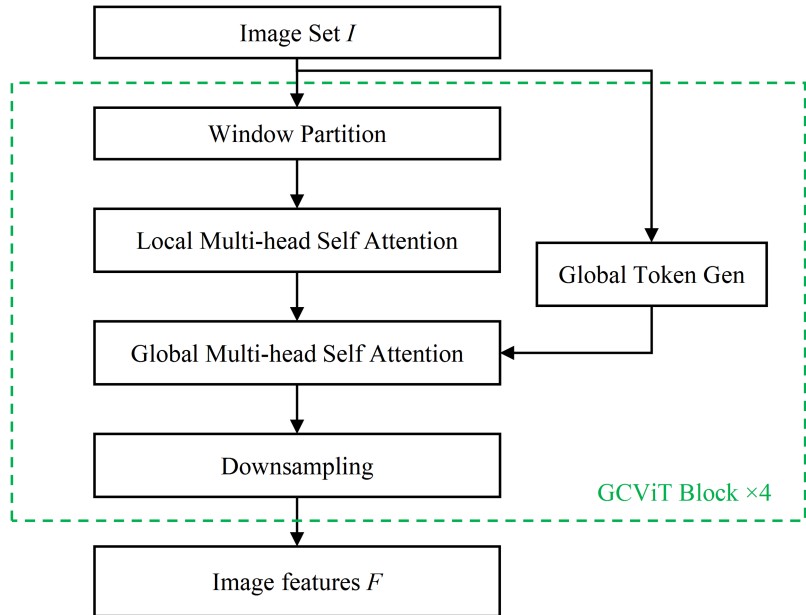

**Figure 2** The image feature extraction module.

Window Partition model, a Local Multi-head Self Attention (MSA) model, a Global MSA model, a Global Token Gen model and a Downsampling model.

The Window Partition model divides the input image data into several windows using partitioning methods (*Liu et al., 2021*), as shown in Fig. 3. In Fig. 3A, the image is divided into four sub-images by windows, and self-attention is computed within each sub-image. In Fig. 3B, the window partition is shifted, resulting in a new division of the image. The two

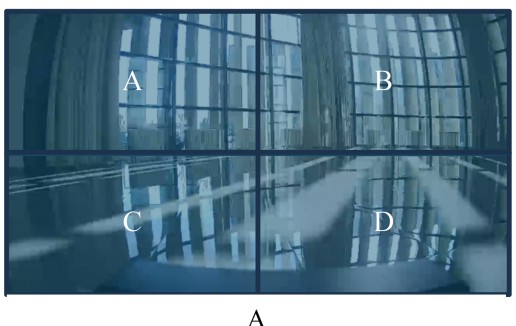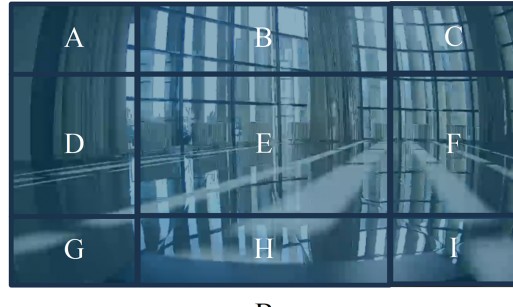

| A | B |
|---|---|

**Figure 3 The window partition model.** (A) Traditional window partitioning. (B) Sliding window partitioning.               

different divisions complement each other by filling in their respective division boundaries, enabling complete attention computation in the boundary area of the partitioned windows. Finally, the original image is divided into $m$ pieces of sub-images $F_{input}$ (where $F_{input} = \{f_1, f_2, f_3, \ldots, f_m\}$).

The Local MSA model calculates attention within pre-segmented sub-images $F_{input}$ and computes attention, as shown in Fig. 4 (*Dosovitskiy et al., 2020*). The $Q$, $K$, and $V$ represent Query vector, Key vector, and Value vector of Attention Block, similarly for $Q'$, $K'$, and $V'$. The Scalar Product refers to the dot product of $Q$ and $K$, while Multiply is for the matrix multiplication calculation. The Softmax function converts the result into probability, and the FC stands for the fully connected layer. The scale is an operator which is multiplied by a scaling factor, corresponding to a constant scaling factor $\sqrt{d}$ in Eq. (1). In addition, the scale serves a regulatory role to ensure that the results remain within an appropriate range.

The Local MSA model is calculated by Eq. (1). The $f_i$ is the $i$-th split sub-image of $F_{input}$, $f_i \in \mathbb{R}^{w_i * h_i * C_i}$, where the $w_i$, $h_i$ and $C_i$ are the width, height, and channel number of the $i$-th split image. The $z$ ($z \in \mathbb{R}^{n*D}$) is the result of multi-head self-attention, while $D$ is the encoding vector dimension, and $n$ is the number of heads in the multi-head attention mechanism (where $n = 8$ in this research). The attention within these sub-images will be calculated separately and merged into the output $F_{Local\ MSA}$ ($F_{Local\ MSA} \in \mathbb{R}^{W*H*C}$, the $W$, $H$ and $C$ are width, height, and channel number of output image respectively).

$$
\begin{cases}
Q, K, V = FC(f_i) \\
z = attention(Q, K, V) = Softmax\left(\dfrac{QK^T}{\sqrt{d}}\right)V \\
f'_{local\ i} = FC(z) \\
F_{Local\ MSA} = \{f'_{local\ 1}, f'_{local\ 2}, f'_{local\ 3}, \ldots, f'_{local\ i}\}
\end{cases}
\tag{1}
$$

The Global MSA model (as shown in Fig. 4) establishes the long-term dependencies by capturing global contextual information, thereby eliminating the need for complex operations. Its inputs are the $F_{Local\ MSA}$ and the Global Token $Q'$ (the output of Global Token Generation model), and its output $F_{Global\ MSA}$ is calculated by Eq. (2).

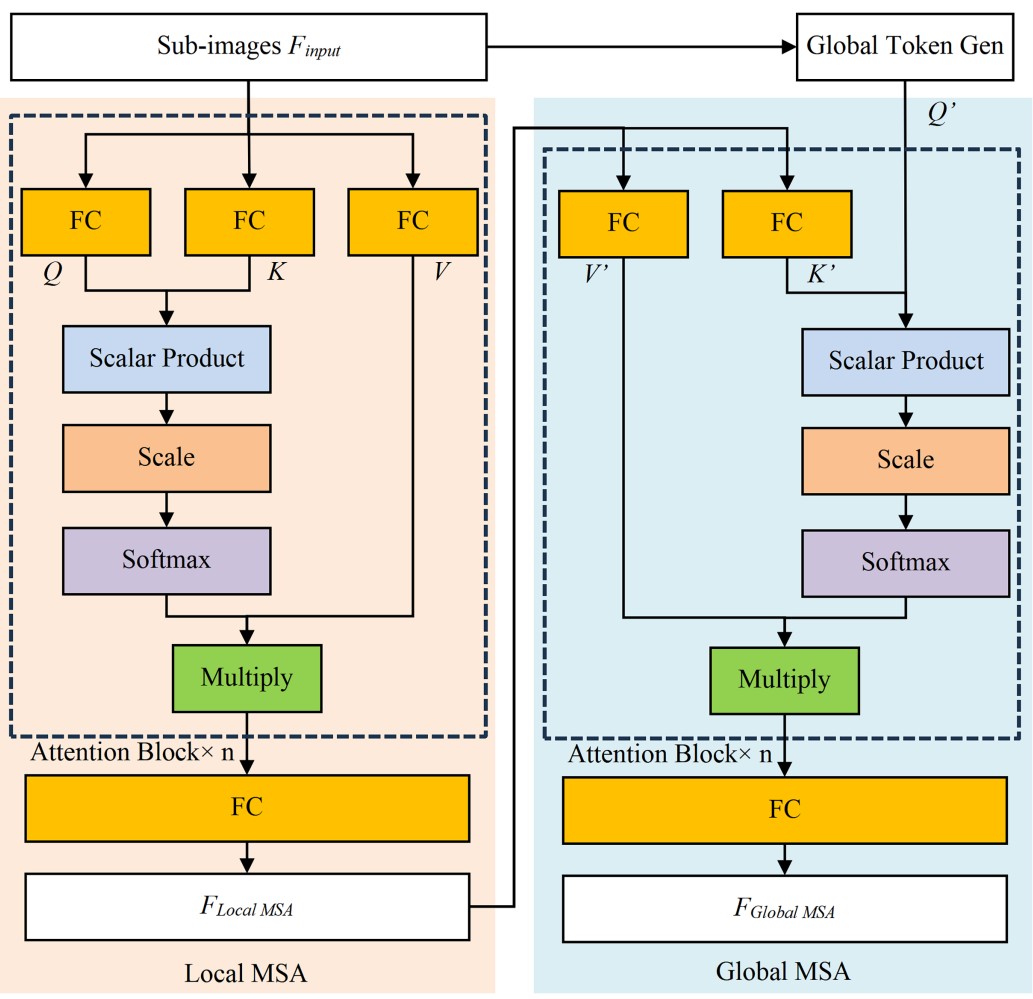

**Figure 4** **The structure diagram of local MSA model and global MSA model.**

$$
\begin{cases}
K', V' = FC\left(f'_i\right) \\
Q' = GlobalTokenGen\left(F_{input}\right) \\
z' = attention(Q', K', V') = Softmax\left(\dfrac{Q'K'T}{\sqrt{d}}\right)V'. \\
f'_{Global\,i} = FC(z') \\
F_{Global\,MSA} = \left\{f'_{Global\,1}, f'_{Global\,2}, f'_{Global\,3}, \ldots, f'_{Global\,i}\right\}
\end{cases}
\tag{2}
$$

The structure of Global Token Gen module is shown in Fig. 5A. The Repeat operation repeats $Q'$ ($Q' \in \mathbb{R}^{wins*C*w_i*h_i}$) to match the $i$-th sub-image number, *wins* represents the number of the $i$-th split image. The Conv2d is a 2D convolutional layer, GERU is the activation function, and MaxPool2d is the maximum pooling layer.

Downsampling module reduces the data dimension, imposes locality bias and cross-channel interaction, as shown in Fig. 5B. The Norm is the normalization layer, while

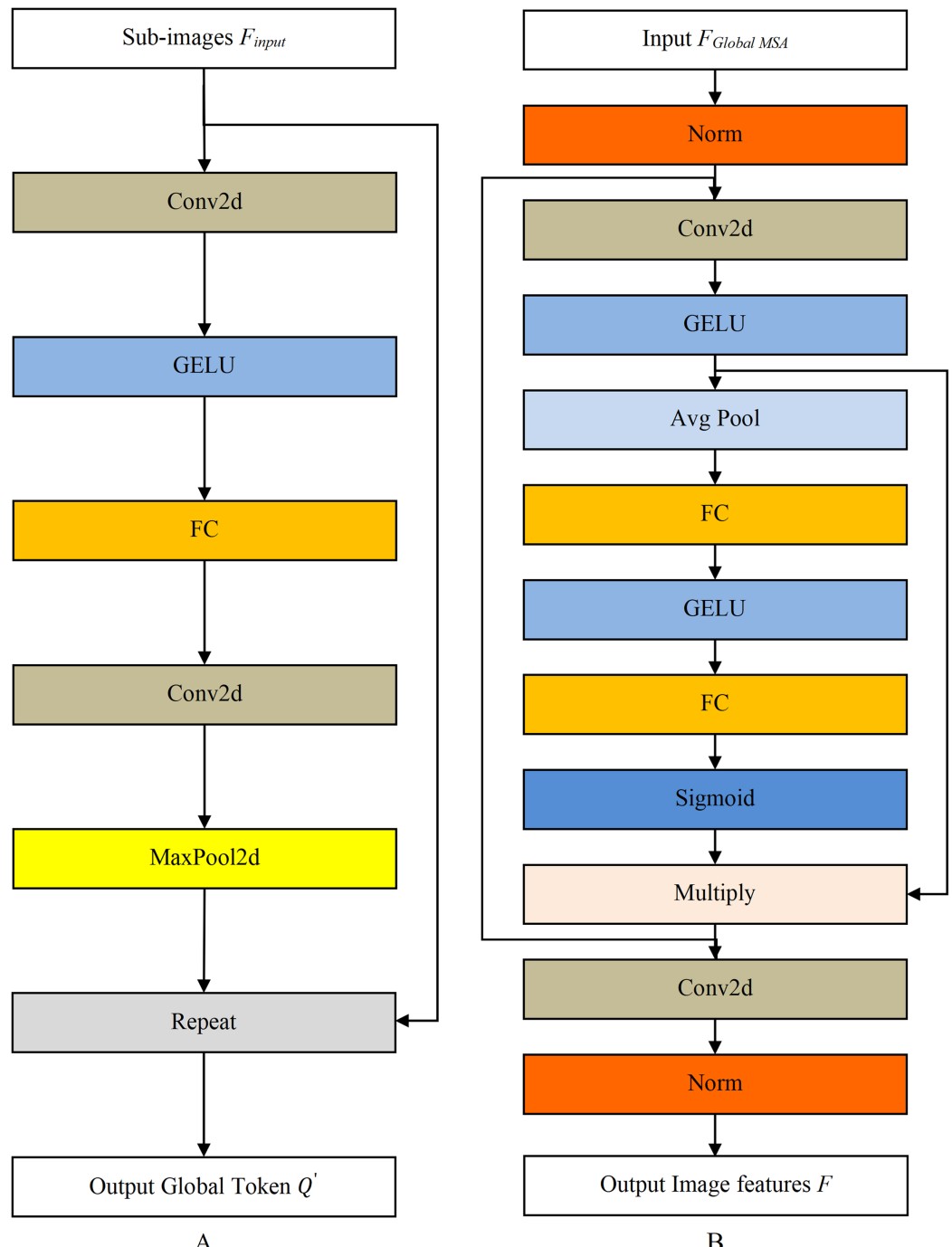

**Figure 5 The structure diagram of Global Token Gen model and Downsampling model.** (A) Global Token Gen model. (B) Downsampling model.

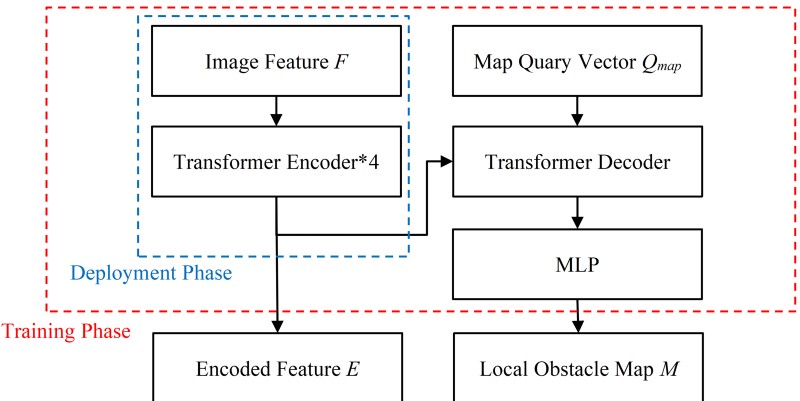

**Figure 6** The structure diagram of Transformer Fusion module.

the Avg Pool and the Sigmoid are the average pooling layer and activation function, respectively.

After four GCViT blocks calculation, the image feature extraction module outputs image feature $F\left(F \in \mathbb{R}^{S*C*\frac{Width}{32}*\frac{Height}{32}}\right)$. Here, $S$ represents the number of images and $C$ represents the number of feature channels, and they are set to 3 and 512 respectively in this research.

## Transformer fusion module

The structure of Transformer Fusion Module is illustrated in Fig. 6. During the training and deployment phase, its Transformer Encoder layer integrates the image features $F$ and generates the fused encoded features $E$ ($E \in \mathbb{R}^{N*D}$, $N = F_w * F_h * I_s$). Where $D$ is the dimension of the Transformer, $F_w$ and $F_h$ are the width and height of the feature map $F$, with values of 7 * 7, and $I_s$ is the number of images. To enhance the interpretability of the network, a Transformer Decoder is utilized during the model training phase to predict the local obstacle map $M$, which is a Bird's Eye View grid probability map. Each grid in $M$ represents a rectangular area of 1 m * 1 m, and the value of the grid indicates the probability of an obstacle existing within that area. The $MLP$ stands for a multilayer perceptron consisting of an input layer, a hidden layer, and an output layer. The $Q_{map}\left(Q_{map} \in \mathbb{R}^{width_m*height_m*D}\right)$ is the map query vector, which is a learnable parameter. The $width_m$ and $height_m$ are the width and height of $M$, and $D$ denotes the dimension of the encoding vector. During the deployment phase, the Transformer Fusion Module only infers the Transformer Encoder layer part without outputting the local obstacle map, thereby saving computational resources and improving the real-time performance.

## Local path prediction module

The structure of Local Path Prediction Module is shown in Fig. 7 (*Chung et al., 2014*). A local path $P$ (where $P \in \mathbb{R}^{2*k}$, and $k$ is 10 in this research) is generated based on the

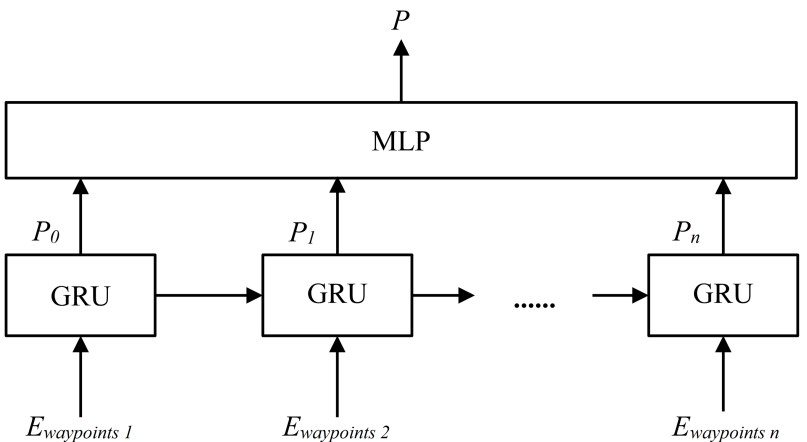

**Figure 7** **The structure diagram of Local Path Prediction Module.**

$E_{waypoints}$, which includes the first 10 vectors in the Encoded Feature $E$. The $GRU$ is Gated Recurrent Unit network and the $MLP$ is the multilayer perceptron.

The local path $P$ is using the vehicle coordinate system, where the positive $y$-axis points towards the front of the vehicle, and the positive $x$-axis points to the left side. The unit length in the vehicle coordinate system is 1 m. The first waypoint of $P$ starts at the center of the vehicle as the origin, with the remaining waypoints are listed in front of the vehicle.

## Controller module

The Controller Module implements vehicle motion control, independently designed from the End-to-End model, enabling cross-platform usage. In this research, the Controller Module employs proportion integration differentiation (PID) controller to compute the vehicle's angular velocity $A$ based on selected waypoints in local path $P$, ensuring that the vehicle maintains path tracking at a constant speed, as shown in Eq. (3). Where $u(t)$ is the output, $e(t)$ is the input, $K_p$, $K_i$ and $K_d$ are the proportions, integral and differential coefficients, respectively.

$$u(e(t)) = K_p * e(t) + K_i \int_0^t e(t)dt + K_d \frac{de(t)}{dt}. \tag{3}$$

The angular velocity $A$ is calculated by Eq. (4). The $p'$ is obtained by averaging the first three waypoints in the local path $P$. The $p'_x$ and $p'_y$ are the components of $p'$ along the $x$ and $y$ axis, respectively.

$$\begin{cases} p' = \dfrac{1}{3}\sum_{t=0}^2 p_t \\ A = u\left(\dfrac{\left(\tan^{-1}\dfrac{p'_x}{p'_y}\right)}{\dfrac{\pi}{2}}\right). \end{cases} \tag{4}$$

## Loss function

The loss function $L$ of the End-to-End autonomous driving model includes two parts: the loss $L_{wp}$ for local path prediction and the loss $L_{map}$ for local obstacle map prediction, as shown in Eq. (5). Here, $P_t$ represents the waypoints in predicted local path; $P'_t$ is the ground truth of the waypoints; $\lambda_{wp}$ and $\lambda_{map}$ are weighted coefficients, both set to 0.5.

$$\begin{cases} L = \lambda_{wp}L_{wp} + \lambda_{map}L_{map} \\ L_{wp} = \sum_{t=1}^{k} \|P_t - P'_t\| \end{cases}. \tag{5}$$

The $L_{map}$ is represented by the object probability prediction loss function $L_{prob}$, which is composed of the positive label function $L_{prob}^1$ and the negative label function $L_{prob}^0$, as shown in Eq. (6). Where $\widehat{M}_{ij}$ and $M_{ij}$ are the true value and predicted value of the probability of the obstacle presence in the $i$-th row and $j$-th column grid; $1_{\left[\widehat{M}_{ij}=0\right]} \in \{0,1\}$ is an indicator function that represents the grid containing obstacle in the label; $C_0$ and $C_1$ are the quantity of grids with obstacles and without obstacles, respectively.

$$\begin{cases} L_{map} = L_{prob} = \left(L_{prob}^0 + L_{prob}^1\right)/2 \\ L_{prob}^0 = \dfrac{1}{C_0}\sum_{i}^{R}\sum_{j}^{R}\left(1_{\left[\widehat{M}_{ij}=0\right]}\left\|\widehat{M}_{ij} - M_{ij}\right\|_1\right) \\ L_{prob}^1 = \dfrac{1}{C_1}\sum_{i}^{R}\sum_{j}^{R}\left(1_{\left[\widehat{M}_{ij}=1\right]}\left\|\widehat{M}_{ij} - M_{ij}\right\|_1\right) \\ C_0 = \sum_{i}^{R}\sum_{j}^{R}1_{\left[\widehat{M}_{ij}=0\right]} \\ C_1 = \sum_{i}^{R}\sum_{j}^{R}1_{\left[\widehat{M}_{ij}=1\right]} \end{cases}. \tag{6}$$

# EXPERIMENT AND RESULTS ANALYSIS

In this research, we utilize the CARLA simulator (version 0.9.10.1) to create a virtual testing field, gather experimental data, and validate the End-to-End autonomous driving model. The model is trained using transfer learning techniques. Urban simulation environments are used for data collection to pre-train the network, followed by collecting appropriate data in a customized temporary road environment for further model training. Ultimately, the efficacy of the model is assessed through both simulation and field testing.

## Experimental dataset collection

The experimental dataset is captured by a rule-based expert model in eight official town maps on the CARLA platform by using the expert agent in Interfuser (*Shao et al., 2023*), as shown in Fig. 8A. Each frame of driving data includes RGB images from left, middle, and right cameras, along with the target path. To enhance the model's adaptability, various weather and lighting conditions are simulated during data collection, including sunny, cloudy, rainy and other scenarios. Besides that, a customized temporary road map, which uses red and blue traffic cones to mark the edges of the road, has been created on the

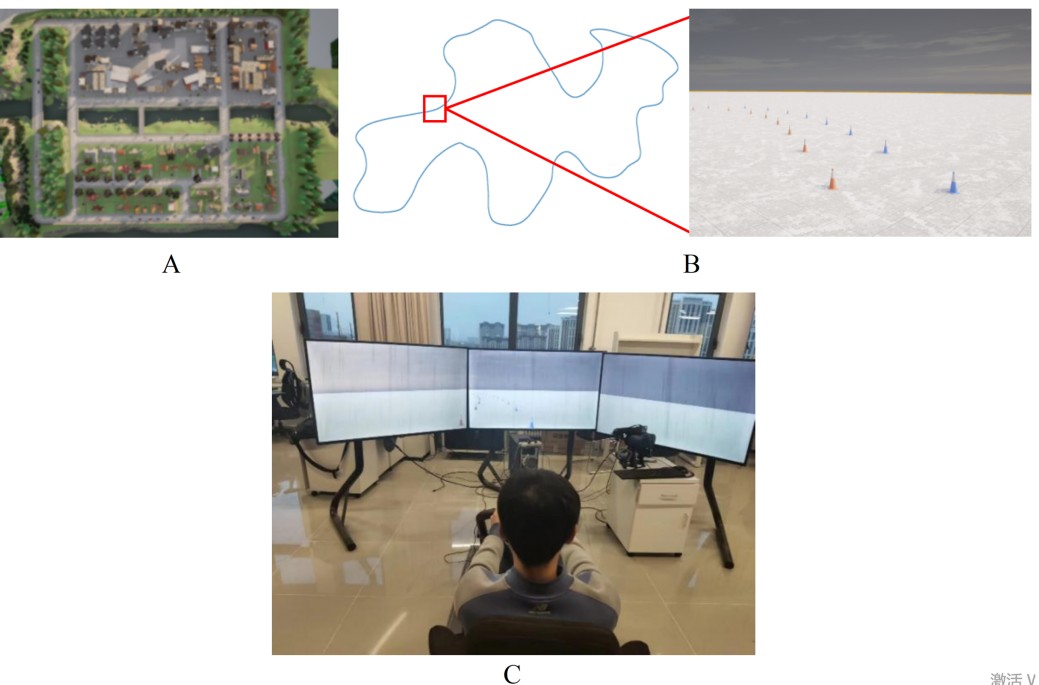

**Figure 8 Data collection in CARLA platform.** (A) Top view of town map. (B) Customized temporary road map. (C) Manual driving platform.

CARLA platform. The length and width of the temporary road are approximately 880 and 3 m, and it incorporates various road conditions such as straight sections, left and right turns, as illustrated in Fig. 8B.

Compared with structured roads in urban environments, paths in temporary road are difficult to generate using rule-based autonomous driving controllers. Therefore, the manual driving annotation is used to generate training data. The manual driving platform consists of a Logitech G29 steering wheel, an accelerator pedal, a brake pedal, a driving seat, a computer and three screens, as shown in Fig. 8C. The driver controls the vehicle along the middle of temporary road under different weather and lighting conditions, and the annotated data will be saved while driving. In addition, the training dataset is also expanded through data augmentation for increasing the data diversity.

## Model performance evaluation in CARLA platform
### Model training and validation

Training the End-to-End autonomous driving model involves two stages: (1) pre-training with dataset generated by expert models in official town map; (2) fine-tuning with dataset captured in temporary road map. The experimental computer configuration included an Intel(R) Xeon(R) Gold 6230R CPU with a frequency of 2.1 GHz, 128 GB of RAM and four NVIDIA GeForce RTX 3090 GPUs with Ubuntu 20.04 Operating System.

During model training, the Adam optimizer is used with a batch size of 16. The learning rate adjustment is shown in Eq. (7), where $\gamma$ the decay factor, $lr_i$ is the learning rate for epoch $i$, *StepSize* is the number of epochs needed for the adjustment, $k$ is a positive integer.

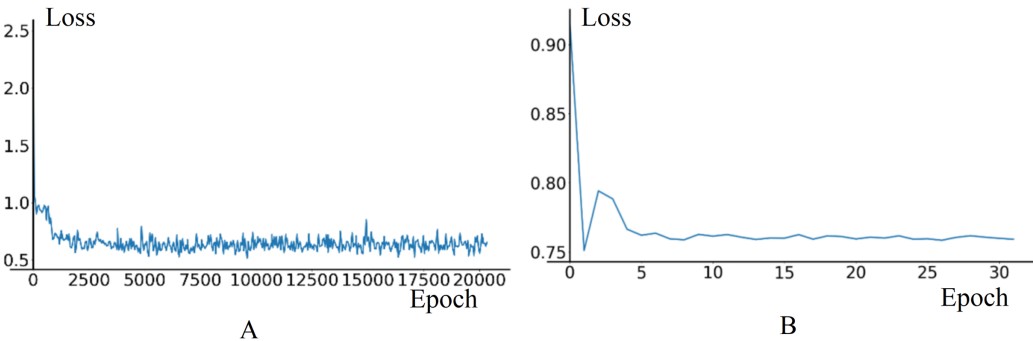

**Figure 9 The loss curves of training and validation in model pre-training phase.** (A) Train loss. (B) Validation loss.

The learning rate will no longer decrease when the minimum learning rate *MinLR* is reached.

$$lr_i = max\left(\left\{ \begin{array}{l} lr_{i-1} * \gamma, \ i = StepSize * k \\ lr_{i-1}, \ i \neq StepSize * k \end{array} \right., \ MinLR \right), \ k \in \mathbb{Z}, \ i > 0. \tag{7}$$

In pre-training phase, the training set consists of 40,256 data frames, while the validation set consists of 11,968 data frames. The initial learning rate for the Image Feature Extraction Module is set to 0.005, and for other modules, it is set to 0.01. The value of $\gamma$ is set to 0.35, *StepSize* is set to 1, *MinLR* is set to 1e−5. The loss curves of training and validation stabilize around 0.4 in training and 0.76 in validation, as shown in Fig. 9.

In fine-tuning phase, the model uses 7,600 training data frames and 1,600 validation data frames generated in the temporary road map. The initial learning rate for the Image Feature Extraction Module is set to 0.0001, with a decay parameter of 0.35 and a *StepSize* of 1. The training and validation results are illustrated in Fig. 10. The loss curve in training phase stabilizes around 0.4. In validation phase, the loss curve exhibits oscillations between the 0th and 5th epochs due to the high learning rate in the initial stage, then, it begins to decrease and ultimately converges around 0.75.

To validate the performance of the GCVit block in Image Feature Extraction Module, it will be replaced by ResNet18, ResNet26 and ResNet50 respectively, as shown in Fig. 11. The result shows that using GCVit as the backbone network performs better than ResNet. Even the initial loss of GCVit is high, it quickly converges after six epochs and reduces to 0.8. The ResNet50 has the largest number of parameters, but its convergence trajectory highly overlaps with that of ResNet18, while ResNet26 performs slightly worse, suggesting that merely increasing the depth of ResNet provides limited improvement.

During the training phase, the End-to-End autonomous driving model outputs a local obstacle map to assist in training. The loss $L_{wp}$ variation is also verified by comparing experiments with and without outputting the local obstacle map, as shown in Fig. 12. According to the experimental result, outputting local obstacle map during training phase can reduce the loss $L_{wp}$, thereby improving model robustness and performance.

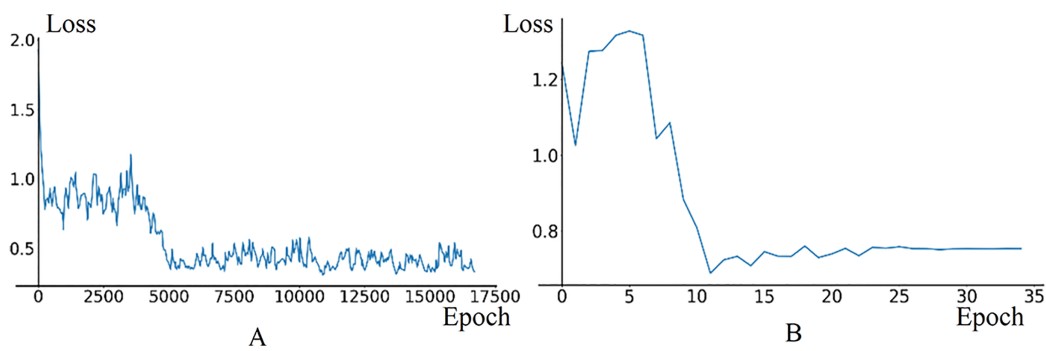

**Figure 10 The loss curves of training and validation in model fine-tuning phase.** (A) Train loss. (B) Validation loss.

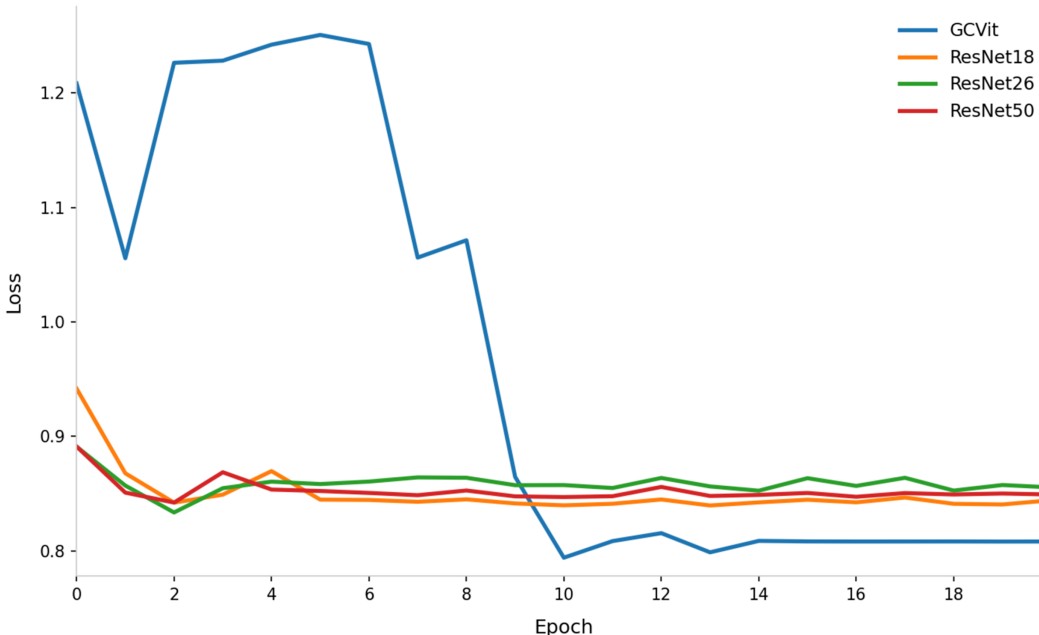

**Figure 11 Comparison of loss curves with different backbone network in Image Feature Extraction Module.**

## Model performance evaluation

Another customized temporary road map has been designed to evaluate the End-to-End model performance, as shown in Fig. 13. The trajectory of manual driving is represented by a blue dashed line, serving as the ground truth, while the trajectory of autonomous driving is marked with a red dashed line. The $K_p$, $K_i$, and $K_d$ parameters of the PID controller are set to 2, 0.15, and 2.3, respectively. Mostly, the automatic driving trajectory and the manual driving trajectory overlap significantly, except in areas with large curves. The average trajectory deviation is 0.689 m, and the maximum deviation is 1.204 m. By further optimizing the parameters of the PID model, the trajectory deviation can be reduced.

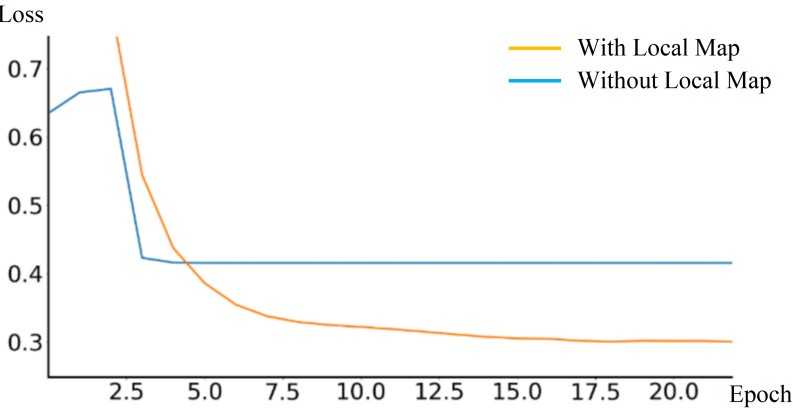

**Figure 12 Comparison of loss curves with and without outputting local obstacle map.**

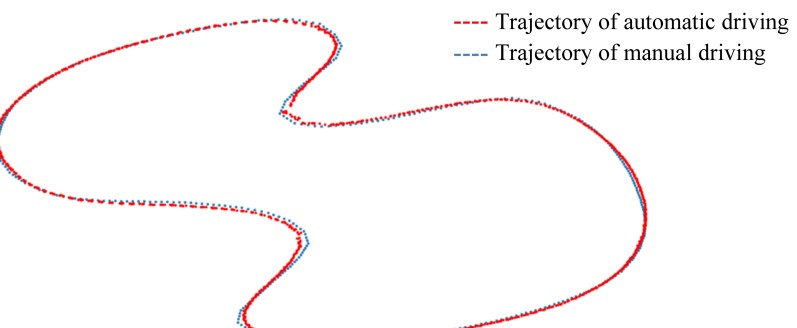

**Figure 13 Deviations in customized temporary road map.**

The output of End-to-End autonomous driving model during temporary road test is shown in Fig. 14. Three RGB images in Fig. 14A are captured from right turn, left turn, and straight road conditions. From the visualization results in Figs. 14B and 14C, it can be seen that the obstacle map labels indicate the traffic cones on both sides of the road, while the manual driving path labels mark the vehicle's driving trajectory. The local obstacle map generated by the End-to-End model uses the identified obstacles as boundaries, and the area within the boundaries is traversable, as shown in Fig. 14D. Based on the local obstacle map, the End-to-End model predicts corresponding local path. Experimental results demonstrate that this path is consistent with the path labels while manual driving, as illustrated in Fig. 14E. Furthermore, the time consumption for a single path planning is approximately 80 ms, which meets the requirements for real-time processing.

Overall, the experimental results on the CARLA platform indicate that our End-to-End model can predict local paths similar to manual driving in temporary road scenarios and achieve the goal of unmanned driving in simulation tests. Next, field experiments with the unmanned ground vehicle (UGV) will be conducted to evaluate the performance of the End-to-End model.

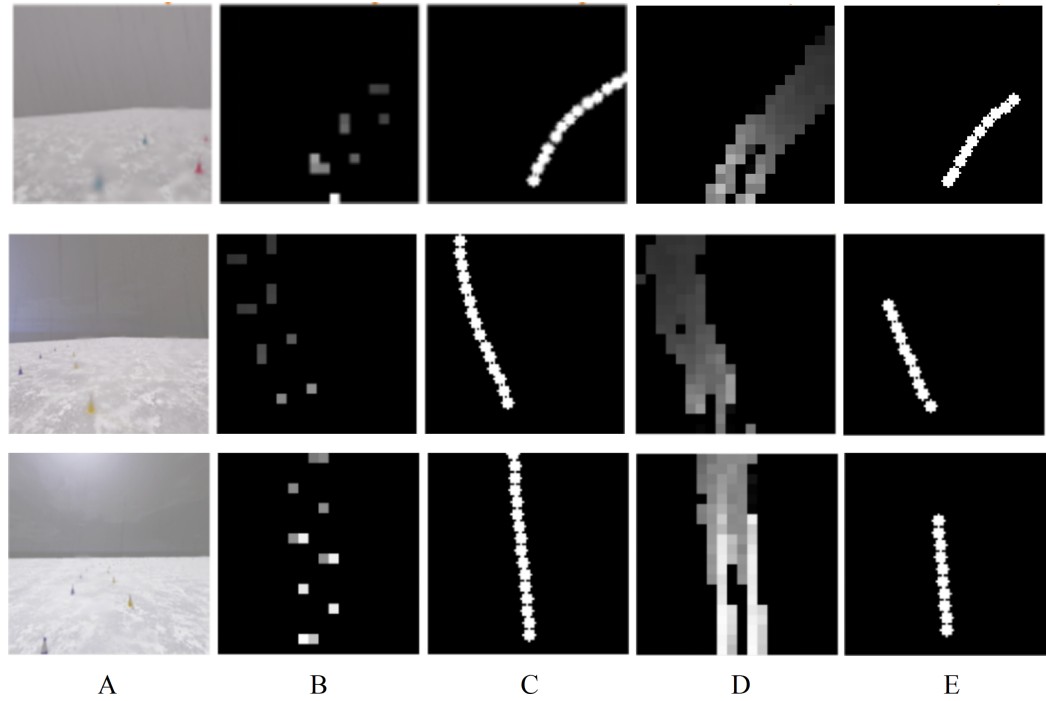

**Figure 14 The output of End-to-End model in CARLA platform test.** (A) RGB road Images. (B) Obstacle map labels. (C) Path labels while manual driving. (D) Predicted local obstacle map. (E) Predicted local path.               

## Model performance evaluation in field experiment

### Experimental devices

The UGV with experimental devices is shown in Fig. 15. The UGV is a XGILEX BUNKER, which has dimensions of 1,023 mm in length, 778 mm in width, and 400 mm in height. It weighs 150 kg, is battery-powered, and uses a tracked drive system. The maximum travel distance is 10 km, and it supports CAN bus communication. The embedded computing platform is the NVIDIA Jetson AGX Xavier, which is responsible for running the End-to-End autonomous driving model. Three HKVISION DS-IPC-T12 cameras are installed for road image acquisition, with an angle of 60° between each camera.

### Performance evaluation in field experiment

In the field experiment, the speed of the UGV is set to 0.5 m/s, the $K_p$, $K_i$, $K_d$ of PID Controller are set to 0.15, 0.75 and 0.3.

Two temporary roads are constructed using blue and red traffic cones. One temporary road is circular, with a width of 1.6 m and an inner radius of 5 m. This road evaluates path tracking performance during continuously left and right turns, with the UGV moving in both clockwise and counterclockwise directions. The other temporary road is S-shaped, consisting of multiple left and right turns, and has a width and length of 1.6 and 25 m, respectively. This road verifies system performance in general road scenarios.

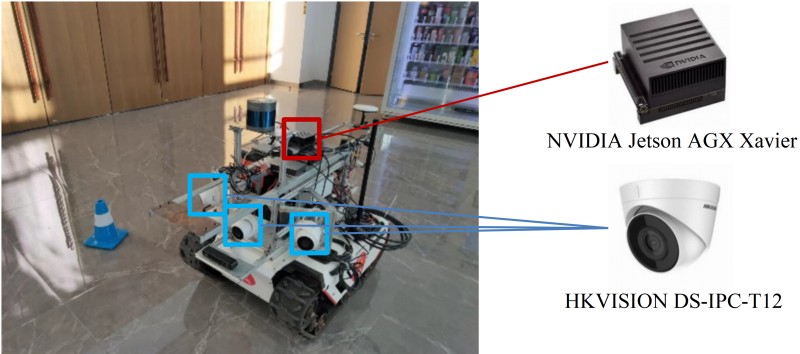

**Figure 15 The UGV and other experimental devices.**

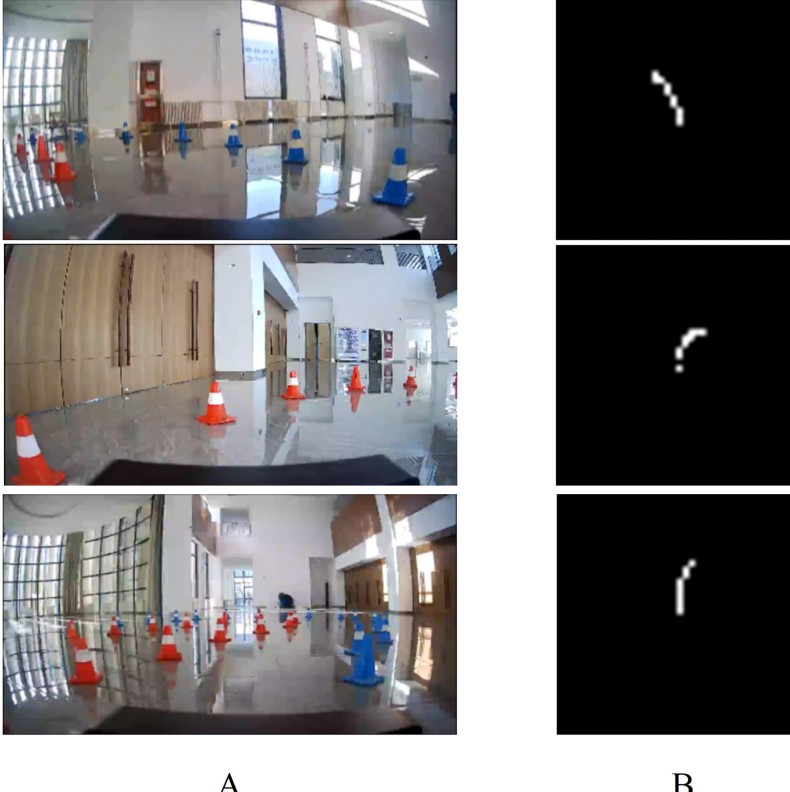

A         B

**Figure 16 The output of End-to-End model in field test.** (A) Road information. (B) Predicted local path. 

As shown in Fig. 16, the variation trend of the predicted local path is consistent with the actual road conditions obtained from the camera images. Although the UGV occasionally collides with traffic cones, it quickly adjusts its direction, returning to the center of the road to complete the temporary road tracking task. The single path planning time is approximately 100 ms, which is 20–30 ms longer than in the simulation experiment. Two reasons account for the increment in time consumption. Firstly, the computational

capability of the NVIDIA Jetson AGX Xavier is inferior to that of the PC running the CARLA platform, leading to longer prediction times. Secondly, data interaction between hardware is necessary in field test, which also contributes to the increased road prediction time. The experiment video can be found at: https://www.bilibili.com/video/BV17w4m1v7c2/?vd_source=50e3da1ec61d287105da0db529781ba1. Field experiment proves that the End-to-End autonomous driving model can extract road features from images captured by three cameras, predict the corresponding path based on road condition, and control the UGV to navigate through temporary roads.

## CONCLUSIONS

This study proposes an End-to-End autonomous driving model for local path planning in a temporary road scenario. The model is based on GCViT to construct a backbone network for image feature extraction, employs a Transformer encoder to fuse the image features, outputs predicted local paths through a GRU network, and generates angular velocity control *via* a PID controller. In both the simulation and field experiments, the End-to-End model is capable of helping the vehicle identify drivable areas on temporary roads, as well as accomplishing path planning and tracking within those temporary roads. In addition, the time taken is around 100 ms, indicating that our End-to-End model meets the real-time processing requirements for autopilot.

Due to the difficulty of parameter tuning in PID algorithms, adjusting its three coupled parameters is challenging. Moreover, fixed-parameter PID controllers struggle to adapt to dynamically changing systems (*e.g.*, when the vehicle load varies). In the future, we will further enhance the self-driving performance by training with larger datasets in more complex environments, and it may be worth considering the use of MPC algorithms to improve the control performance.

### Funding
This work was supported by the Foundation Strengthening Program Fund Project (2021JCJQJJ0022), the National Natural Science Foundation of China (52175074), and the Dynamic Experimental System for Integrated Transmission Bench (S19261000). The funders had no role in study design, data collection and analysis, decision to publish, or preparation of the manuscript.

### Grant Disclosures
The following grant information was disclosed by the authors:
Foundation Strengthening Program Fund Project: 2021JCJQJJ0022.
National Natural Science Foundation of China: 52175074.
Dynamic Experimental System for Integrated Transmission Bench: S19261000.

### Competing Interests
The authors declare that they have no competing interests

## Author Contributions

- Qinghua Su conceived and designed the experiments, prepared figures and/or tables, authored or reviewed drafts of the article, and approved the final draft.
- Min Xie conceived and designed the experiments, performed the experiments, performed the computation work, prepared figures and/or tables, authored or reviewed drafts of the article, and approved the final draft.
- Liyong Wang conceived and designed the experiments, authored or reviewed drafts of the article, and approved the final draft.
- Yue Song analyzed the data, prepared figures and/or tables, and approved the final draft.
- Ao Cui performed the experiments, performed the computation work, prepared figures and/or tables, and approved the final draft.
- Zhihao Xie performed the experiments, authored or reviewed drafts of the article, and approved the final draft.

## Data Availability

The data is available at GitHub:

- https://github.com/cuiao66/ConeRouteNet.git
- MichstaBe. (2025). cuiao66/ConeRouteNet: ConeRouteNet (v1.0). Zenodo. https://doi.org/10.5281/zenodo.16756879

The code is available in the Supplemental Files.

## Supplemental Information

Supplemental information for this article can be found online at http://dx.doi.org/10.7717/peerj-cs.3152#supplemental-information.

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
