# Peer review of "An End-to-End autonomous driving model based on visual perception for temporary roads"

_PeerJ Computer Science, doi:10.7717/peerj-cs.3152_

## Round 0.1 · original submission · Major Revisions

Please consider the comments carefully and improve the article accordingly. Then the revised version will be evaluated again.

**Language Note:** The review process has identified that the English language must be improved. PeerJ can provide language editing services - please contact us at [email protected] for pricing (be sure to provide your manuscript number and title). Alternatively, you should make your own arrangements to improve the language quality and provide details in your response letter. – PeerJ Staff

Reviewer 1 ·

Basic reporting

The article is clearly written and professionally structured, including all necessary figures, data, and methods. There are some minor grammar issues and awkward phrasing (e.g., Lines 43–45), but they do not hinder understanding. A light language polish would enhance readability.

Experimental design

The research question is well-defined and relevant. The use of GCViT, Transformer, and GRU in the proposed pipeline is technically sound. Dataset collection in CARLA and real-world UGV testing demonstrates rigor. Methods are reproducible and described in sufficient detail.

Validity of the findings

Results are robust and clearly presented. The addition of the local obstacle map improves training performance, and the average trajectory deviation and inference time are well within practical bounds. Conclusions follow logically from the data.

Additional comments

-- Minor grammar/language edits throughout.

--A brief discussion of limitations (e.g., PID control tuning, generalization) would strengthen the conclusion.

--Some figures (e.g., Figures 5–6) would benefit from clearer labels.

Reviewer 2 ·

Basic reporting

see 4. Additional comments

Experimental design

see 4. Additional comments

Validity of the findings

see 4. Additional comments

Additional comments

The authors proposed an end-to-end model to drive a vehicle in simulation world using CARLA and real-world using UGV. The presentation of the paper is easy to follow, however, there are some revision needed to improve the quality of the research.
1. The authors should conduct comparative study with some latest models to better understand the model performance. There are plenty of works in the field of end-to-end autonomous driving.
2. In addition to point 1, the authors should also conduct ablation studies by creating some variants of their proposed model. The variants could be made by changing some parts of the network architecture to understand the influence of the changed part.
3. Since it is based on imitation learning where an expert provides a lot of driving records for training, the authors should elaborate on the dataset information, multi-task loss functions, training configurations etc.
4. Since the authors provide real-world case, how do the authors handle the sim2real gap loss? It must be explained.
5. It would be better if the authors also provide offline test on public driving dataset like nuscene, kitti, etc.

·

Basic reporting

1. In Line 18, please elaborate “urgent control requirements”. Providing some examples here will help the readers better understand the motivation behind this research work
2. Please add the source code github link “https://github.com/cuiao66/ConeRouteNet” in the manuscript
3. Consider adding a README file describing how the model was trained, how to set up the simulation environment, and any other relevant details for reproduction
4. In Line 42, the following phrase “Compared with achieved” is unclear. What is it being compared with?
5. The statements in Lines 40-45 have been repeated again in Lines 46-49.
6. In Line 32, please remove the phrase “Add your abstract here.”
7. In Line 157, please define “n” in the equation
8. In Line 230, please explain what is “PID”
9. In Line 292, please describe why sampling is done in the interval [0.4, 1.6]
10. In Line 293, equation 12 - please explain why the constant 127 was chosen
11. The following grammar and typo changes need to be done
a. In Line 43, “the field of autonomous establishing a direct mapping driving due to their from sensor high accuracy data to decision-making” this statement is phrased incorrectly
b. Line 45, change “The approach of can effectively reduce uncertainty” to “The approach of effectively reducing uncertain”
c. In the title of Figure 16, remove unnecessary spaces in the following words “D eviations”, “c ustomized”, “r oad”. Similar issues in other figure titles as well. Please correct all of them
d. In equations 4,5,6 the subscript for E has “waypointst”. Is this a typo and should this rather be “waypoint”?
e. Line 71, change “End to end” to “End-to-end”
f. Lines 156-157, Line 169, change “splited” to “split”
g. Line 142 and Line 158, change “subimages” to ”sub-images”
h. Line 142, no closing bracket for “(where”
i. Line 157, change “selfattention” to “self-attention”
j. Line 163, change “input” to “inputs”
k. Line 167, change “structure Global” to “structure of Global”
l. Line 168, change “matching to the” to “match the”- If the suggested change isn’t correct please rephrase the sentence
m. Line 169, change “GERU is activation” to “GERU is the activation”
n. Line 170, change “MaxPool2d is maximum” to “MaxPool2d is the maximum”
o. Line 232, change “u(t) is output, e(t) is input” to “u(t) is the output, e(t) is the input”
p. Line 256, change “pretrain” to “pre-train”
q. Line 275, change “platform is consists” to “platform consists”
r. Line 326, change “epoch” to “epochs”
12. The following formatting changes should be considered by the authors for improved readability
a. Leave a line gap between sections in the manuscript - for example, after line 32 and before line 33 ('Introduction'), as well as between lines 49 and 50.
b. Line 345, change “Lwp” to the mathematical version i.e., L with subscript of wp
c. Line 143, change “model(“ to “model (“
d. Line 76, change “control(Bojarski” to “control (Bojarski”
e. Line 88, change “intersections(“ to “intersections (“
f. Line 97, change “2017;Wu” to “2017; Wu”
g. Line 247, “equation(11)” to “equation (11)”
h. Please follow a consistent spacing when mentioning figures. For example Line 285 has “Fig. 12 (a)” but in line 289 there is no space “Fig.12(c)”

Experimental design

1. Line 60, can you describe what constitutes a temporary road? In your research, what kind of temporary roads were considered and what was out of scope? Providing a clearer description here would be helpful
2. In Line 125, please also specify the field of view of each camera

Validity of the findings

1. While trajectory deviation is okay for validating model performance, it might not be sufficient for safety. Please consider validating your model in a simulated environment to check if it navigates around obstacles, stops for pedestrians and changes trajectories according to the agents in the scene
2. For model performance evaluation, please consider testing how well can the model generalize to unknown temporary road structures. For example, if the cones were replaced with traffic barriers would the model still detect and plan trajectories properly?

Additional comments

I commend the authors for using a robust approach to validate their autonomous model through both simulation and structured testing. The use of a real vehicle for performance evaluation shows that the author’s approach is practical and is truly an end-to-end solution.

---

## Round 0.2 · accepted · Accept

Thanks to the authors for their efforts to improve the work. This version has addressed the reviewers' concerns successfully. It can be accepted. Congrats!

·

Basic reporting

no comment

Experimental design

no comment

Validity of the findings

no comment